# Marked Range Regression and Possible Alteration of Distribution of the Dupont's Lark *Chersophilus duponti* in Tunisia: Conservation Consequences of Vanishing Alfa Grass *Stipa tenacissima* Steppes in North Africa

**Javier Viñuela** [1],*[ID]**, Jesus T. García** [1][ID]** and Francisco Suárez** [2]

1 Instituto de Investigación en Recursos Cinegéticos (IREC, CSIC-UCLM-JCCM), Ronda de Toledo 12, E-13071 Ciudad Real, Spain; jesusgarcia.irec@gmail.com
2 Departamento de Ecología, Universidad Autónoma, Darwin 10, E-28049 Madrid, Spain
* Correspondence: javier.vinuela@uclm.es

**Abstract:** The effect of global warming and desertification on bird populations of semi-arid North African ecosystems has been little studied, although ecosystems in those areas are suffering dramatic changes. Dupont's lark is one of the most endangered passerines in Europe, but it is also considered scarce in North Africa, where its range and numbers are only well known for Morocco. We analyzed the current distribution and population size of Dupont's lark in Tunisia and compared the current figures with older data. To assess the presence of the species in the patches of adequate habitat that we found, we elicited territorial calls by broadcasting the males' territorial songs during early breeding season (N = 123, ≈40 h). Fieldwork (45 persons/day) and analysis of satellite images were combined to determine the current minimal extent of occurrence and area of occupancy, following IUCN definitions. In the only habitat where we found the species (well-preserved pure or mixed alfa patches in the Feriana-Kasserine region), the extent of occurrence in and effective area of occupancy were small (56.3 and 33.2 km$^2$, respectively), particularly when compared with previous estimates (presence of the species in adequate habitat over ca. 89,000 km$^2$). The species has not been detected at all in a large area in the southern part of its potential range, where additional surveys are urgently needed to locate possible remnant small and isolated populations that could persist, as suggested by two recent records of the species there. The breeding population of Dupont's lark in Tunisia is estimated to be fewer than 600 song birds (335–577). The drastic reduction of range and numbers of this species must have been caused by the disappearance or degradation of alfa grasslands due to agricultural development, overgrazing, and increased aridity.

**Keywords:** birds; larks; North Africa; conservation; global change; population trends; steppe

## 1. Introduction

There is growing concern about the possible effects of global warming on animal populations [1,2], particularly in areas already exposed to desertification risk, such as the western Mediterranean basin, where available information in this respect is relatively scarce [3]. The arid and semi-arid ecosystems of Maghreb have suffered deep ecological changes at the landscape level during the last decades, including a dramatic loss of vegetation cover and a simultaneous advance of Saharan ecosystems northwards, a situation that could be impaired by ongoing global warming [4,5]. The loss of steppe vegetation in northern Africa has been estimated to be approximately 50% in the second half of the twentieth century [5].

Resident species with limited dispersal ability and strongly associated to low and sparse shrub vegetation, such as the Dupont's lark *Chersophilus duponti* [6–8], may be particularly affected by these drastic changes in vegetation cover [9]. This is one of the most endangered passerines in Europe, with its actual range in the continent being limited

to Spain [10]. The population of Dupont´s lark in Spain has been recently estimated to be no more than 1400–1500 territorial pairs, with a yearly rate of 3.9% population decline. Modeling work suggests a probability of extinction higher than 80% in just two decades, so it has been considered as Endangered at the national level since 2004 [6,11–14]. The main conservation problems of the species in Spain are habitat loss driven by agricultural expansion, reforestation schemes, wind farm construction, rural abandonment, and other land use changes; it is also a species highly sensitive to climate change [14–17].

The range of Dupont´s lark extends into northern Africa, where two subspecies have been described, mainly on the basis of morphological differences [10,18]. The nominate subspecies (*C. d. duponti*) is thought to occur in Spain, Morocco, and northern Algeria and Tunisia, while *C. d. margaritae* is thought to occupy southern areas at the edge of the desert in Algeria and Tunisia, as well as a coastal fringe in Libya and Egypt [10,19] (see Figure S1). However, the distribution, range, and population size of the species in northern Africa have only been reported for Morocco (an estimated overall population of about 15.000 pairs) [7], but little is known about its status in other geographic areas within the Maghreb [10,19,20]. Despite this scarcity of information about the real status of the species in most countries of northern Africa, it has recently been upgraded to Near Threatened at the world level [21] and to Vulnerable since 2020 [19], reflecting the concern that main populations would have declined along recent decades (e.g., [22] for Morocco and [18] for Tunisia).

In Tunisia, the species is considered sedentary and thought to occupy a wide strip between 33°00′ and 34°20′ N, where apparently both subspecies coexist, although it is not clear to what point their ranges overlap or not [18]. The nominate subspecies is said to occupy the "central plains", while *C. d. margaritae* is said to be restricted to areas "south of Gabes and the Chotts" [18] (Figure 1). Recent work has shown that there is a clear genetic differentiation between Tunisian populations and those of Morocco and Spain, currently reflected in clear morphological differences as well, apparently the same as those detected by classical taxonomy [8]; however, both subspecies would be currently suffering ongoing hybridization [23]. Thus, although data are lacking from Algerian, Libyan, and Egyptian populations, it seems that in Tunisia there is a clearly distinct genetic variant that does not exist in Spain or Morocco. However, little is known about the distribution and status of this unique taxon, which is a relevant additional factor to improve our knowledge about habitat availability, distribution, and status of this species in Tunisia, even though it may not be clear which subspecies is being sampled. There are also clear indications that the potential range and numbers of Dupont's lark in Tunisia may have been greatly reduced during the last decades due to extensive habitat loss caused by overgrazing by livestock and agricultural development affecting areas with natural vegetation [18]. However, information about the recent distribution of the species in Tunisia is scarce, and the estimated current distribution is based on data from ornithological expeditions during the 1950s [24]. More recent information seems to be limited to a few isolated and disperse sightings (see references in [18]).

One of the main habitat requirements of Dupont´s lark is adequate vegetation cover. While this species may be found in Spain in a variety of shrublands [25], in northern Africa, alfa grass steppes have overwhelming importance, harboring 91% of the estimated population of Dupont's lark in Morocco [7]. Alfa grasslands occupy a narrow strip along the northern edge of typical Saharan ecosystems and are even considered the ecological landmark limiting the desert [26]. This habitat may thus be particularly sensitive to overgrazing, cultivation, the advance of desertification [4,5,27], and to climate change, particularly in the most arid areas of the Mediterranean basin, as suggested by the observed and predicted sharp increases in temperature, summer droughts, and aridity [3,28].

Reports from fifteen years ago [7] on the degradation of large areas of alfa grasslands in the southern edge of the Dupont´s lark range in eastern Morocco and concomitant declines of their populations have been recently confirmed [23]. Therefore, updating information about the Dupont's lark distribution, population size, and trend in Tunisia, as well as

the assessment of current conservation problems in this region, would be important for conservation purposes.

The aims of this study were to (1) determine the current distribution of the Dupont's lark core areas in Tunisia, (2) sample all potential current habitats for the species to provide an unbiased estimate of its distribution and population size, and (3) provide an updated assessment of the main conservation problems in this area.

## 2. Materials and Methods

The whole distribution range of the species in Tunisia according to [18] was prospected during the second half of January and first half of February 2007, with the exception of coastal areas around Gabes (Figure 1A). Basically, the area sampled extends over a 200 km strip (covering ca. 89,000 km$^2$) with at least four well-defined vegetation types and different degrees of habitat alteration by human activities (Figure 1B): (1) NW, the Feriana-Kasserine region, dominated by alfa grasslands; (2) extensive shrub patches (and absence of alfa grass), close to Gafsa city; (3) SE, dominated by mixed shrubs, with some remains of alfa grass and *Lygeum sparteum* or *Hammada scoparia* patches at Tataouine and Ben Guerdane regions, where some recent Dupont's lark records exists; and (4) the halophytic shrub formations around Chott El Jerid, mainly in the Kebili and Douz region.

To evaluate the distribution of the species across this range, although we primarily searched for alfa grass and *Artemisia* spp. shrub in lime or pebbly soils (as described by [24,29]), we also searched in other vegetation types offering the minimum vegetation cover and height needed by the species, according to the available information for the Spanish and Moroccan populations [7,25]. Thus, we searched for birds in halophytic shrubs (*Salsola* spp. and *Suaeda* spp.), and mixed-species xerophytic vegetation (e.g., with *Lycium intrincatum*, *Hammada scoparia*, *Astragalus armatus*, and *Lygeum sparteum*). We also sampled some areas with relict alfa grass patches, but sandy enough to maintain sand-loving vegetation (present mainly in the southern part of the study area, near Tataouine and Ben Guerdane). In each area, we tried to select sampling sites with the best habitat in relation to vegetation cover and height to maximize the probability of detection of this elusive species.

When large patches of good habitat were found, we distributed several sampling points across the patch to obtain an unbiased sampling proportional to the area covered by adequate habitat (between 2 and 40, depending on patch size and the area covered by good habitat inside each patch; Figure 1B). As our main aim was to establish the core areas for the species, habitat patches smaller than 1 km$^2$ and isolated from main habitat patches (>30 km distance) were not prospected (see [7] for a similar approach in Morocco). Only areas of flat or rolling terrain were prospected since the species is absent from slopes steeper than 15%, at least in Spain and Morocco [7,30]. Surveying effort was 45 persons/day. Presence of the species was identified following the procedure detailed in [7] for Morocco, allowing a direct comparison of results. Shortly thereafter, we elicited territorial calls by broadcasting the male territorial songs and calls and noted visual or auditory detection of individuals over a 20 min duration ("playback trials"). The alarm calls of Dupont's lark can be detected year round [31], which enabled us to detect presence by playback trials at any time (see [7]). The position of each detected bird (using a Global Position System, GPS, error ± 10 m) and the habitat type were recorded as in [7]. The dates of the survey were near the start of the breeding season in North Africa (February, with first clutches in March; [8,18]), so we expected as good a response to the broadcasting trials by resident territorial birds as we found in Morocco for similar dates [7,8]. As reported below, we only detected the species in the Feriana-Kasserine region, where we developed an intensive sampling (Figure 1C, Table 1) with the aim of capturing birds to obtain information about the taxonomic identity of individuals [23].

**Table 1.** Percentages of song broadcasting trials with positive response by Dupont's larks in different regions and habitat types. Number of trials in brackets.

| Habitat | Gafsa | Feriana-Kasserine | Tataouine | Kebili | Total |
|---|---|---|---|---|---|
| Alfa grass | - | 41.3 (63) | - | 0.0 (5) | 38.2 (68) |
| Salsola shrub | - | - | - | 0.0 (10) | 0.0 (10) |
| Hammada shrub | - | 0.0 (4) | 0.0 (18) | 0.0 (2) | 0.0 (24) |
| Other shrubs | 0.0 (8) | - | 0.0 (12) | 0.0 (1) | 0.0 (21) |
| Total | 0.0 (8) | 38.8 (67) | 0.0 (30) | 0.0 (18) | 21.1 (123) |

To estimate the whole extent of suitable habitat and the potential distribution area for the species in the Feriana-Kasserine region, the boundaries of the different habitat type patches were accurately determined by means of itineraries by car and/or by scanning the area with binoculars and telescopes (20–60 × 80), and then they were geo-referenced (by GPS). The boundaries of these units were then confirmed and, in each case, extended over nearby areas (maximum 30 km) by means of visual similarity analysis on satellite images (three Landsat ETM + bands, GeoCoverTM, year 2000, pixel size 14.2 × 14.2 m) (Figure 1C). All geo-referenced patches and bird locations were incorporated into a Geographical Information System (Arcview 3.2) for automatic calculations. Following the IUCN criteria [32], the 'extent of occurrence' was measured as the area inside a minimum convex polygon (the smallest polygon in which no internal angle exceeds 180°) containing all the sites of recorded or inferred occurrence (i.e., all delimited patches with suitable habitat) and subtracting discontinuities of unsuitable habitat within the overall distribution of the species. We considered as 'patches with suitable habitat' all those in which the species was recorded, as well as all those with similar vegetation within a range of 30 km to a recorded individual. The 'area of occupancy' was considered to be the sum of the areas covered by all those patches in which the species was recorded (e.g., exactly the same methodology applied in Morocco, see [7]).

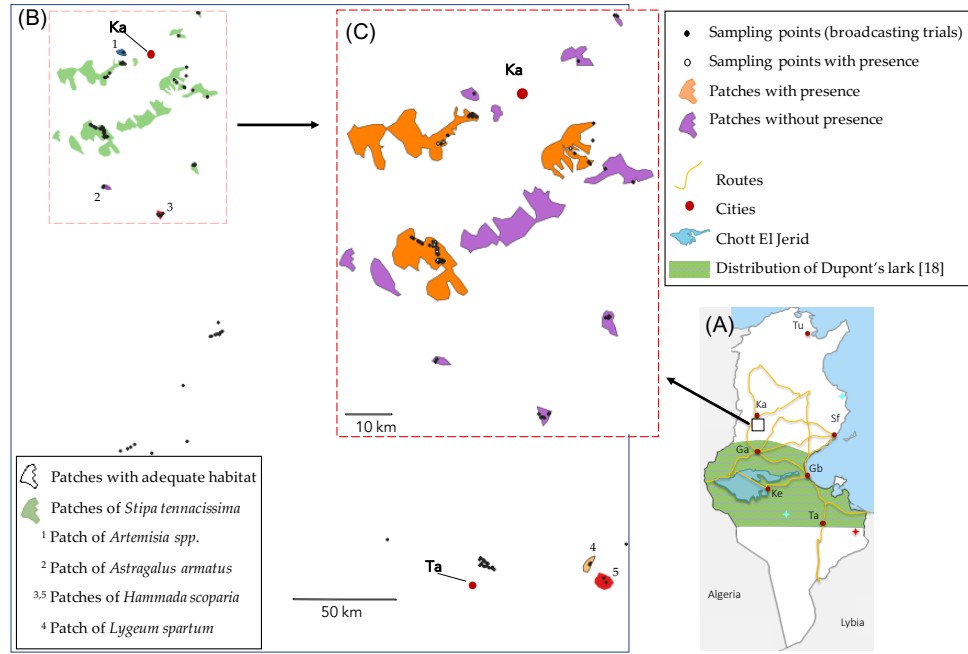

**Figure 1.** Detailed summary of the Dupont´s lark survey in Tunisia. (**A**) Map of Tunisia showing the breeding distribution of Dupont's lark according to [18]. (Ka = Kasserine; Ga = Gafsa; Gb = Gabes; Ke = Kebili; Ta = Tataouine; Sf = Sfax; Tu = Tunis). The only recent record of the species (internet search) is marked as a red star (Sidi Toui National Park, [33]). Recent (after 2007)

observations (E-Bird Platform) are indicated as blue stars (see details in Table 2 and main text). The unfilled square represents the area in which the species was found (see panel C). (**B**) Location of all sampling points and main habitat patches from Kasserine in NW to Tatatouine in SE. Points outside polygons correspond to halophytic shrub formations around Chott El Jerid (with mainly *Salsola* spp. and *Sueda* spp.) or to other mixed-shrub formations with no clear dominant species. (**C**) Detail of sampling in the area with detected presence of the species(Feriana-Kasserine).

When fieldwork was performed, during late winter, males were not singing in a regular way yet. For this reason, we did not use any survey method based on census of singing males that are commonly used in similar studies [6,7]. Instead, we considered two different methods to estimate density. The first one was based on intensive sampling of two habitat patches (76 and 128 ha) by a network of song trials distributed as regularly as possible within the study area, keeping a minimum distance of 500 m between them (n = 7 and 13 trials). In these areas, we tried to capture all males attracted by the broadcasted song (see details of the trapping method in [7]), and they were marked with color rings to avoid multiple records of the same individuals. This method would provide a conservative minimum estimate since we could not be sure that all males present in the area responded to the calls. The second method was based on the fact that the frequency of response by wild birds to song broadcasting is a good indicator of population density in a given area [7]. Then, for each habitat type, we calculated the frequency of response and compared these data with the equivalent for Morocco obtained at the same time of year and in similar habitats in areas with known density [7]. Density was estimated as the ratio between response frequency in each habitat for Tunisia and Morocco, multiplied by population density in Morocco in the same habitat type. Densities estimated in Morocco may be considered as reliable given the large area sampled [7]. Maximum, minimum, and average population sizes of Dupont's lark were calculated as the product of the estimated densities (and their 95% probability upper and lower confidence limits according to the variance obtained in Morocco) by estimated extent of occurrence. To analyze possible differences in detection rates between habitats or regions, we used logistic regressions. All analyses were made using an SPSS v.12 statistical package.

Additionally, to complete our survey with any other recent observations of the species reported in Tunisia, we performed an internet search of Dupont's lark records in that country during the last 20 years by using the search term "(Dupont's lark OR Chersophilus duponti) AND Tunisia". We also compiled observations of the species across all ranges uploaded to the E-Bird platform (https://ebird.org/home, access on 3 January 2023). To avoid the overcounting of observations, we considered sightings of the species by several persons on the same point and date as the same birds watched simultaneously by a team of birders (a circumstance confirmed for several cases in Spain).

## 3. Results

The total number of playback trials was 123, with a total trial time of 41 h, during which 31 birds were detected and their locations geo-referenced. The distribution of trials among regions and vegetation types is summarized in Table 1 and Table S1. The species was detected only in three extensive alfa patches of the Feriana-Kasserine region (Figure 1C), while all trials had negative results in the other regions or habitat types (Table 1). The number of detected birds varied significantly between regions and vegetation types (logistic regressions, $p < 0.001$ in both cases).

It must be noted that the only area where the birds were detected extends the estimated range of the species about 60 km to the north, while the birds have not been found in the rest of the large area considered to be the potential range of the species (Figure 1; [18]). The extent of occurrence was 56.3 km$^2$ with 11 alfa grass patches (mean $\pm$ sd of alfa patch, 5.1 $\pm$ 4.9 km$^2$; Figure 1C). Overall area of occupancy covered 33.2 km$^2$ (58.9% of the extent of occurrence) and was composed of 3 large alfa grass patches (11.1 $\pm$ 3.0 km$^2$; Figure 1). Song trials were performed in 63.6% of the patches of the extent of occurrence, corresponding to 69.9% of its surface.

Densities estimated in the two intensively sampled patches were 6.57 and 6.25 singing birds/km$^2$ (mean, 6.41 birds/km$^2$). Applying this estimated density to the extent of occurrence described previously, we would estimate an overall population of only 361 males. The percentage of positive trials in Tunisian alfa patches was only slightly smaller than that obtained in Morocco in the same habitat and time frame, and the difference was not statistically significant (Tunisia, 41.3%, n = 63; Morocco, 42.1%, n = 38; Chi2 = 0.067, df = 1, *p* = 0.9). From these data, and correcting for the latter difference in response frequencies (e.g., assuming that density in Tunisia would be proportionally lower), we would estimate Dupont's lark density in Tunisian alfa steppes to be 8.85 ± 5.04 territorial males/km$^2$ with 95% confidence intervals maximum and minimum values of 10.83 and 6.29 territorial males/km$^2$, respectively, (9.02 ± 5.14 territorial males/km$^2$ in Morocco [7]). The overall population estimated with this method would be 498 territorial males (maximum, 577 males; minimum, 335 males).

Our internet search produced just one record of Dupont´s lark (likely the *margaritae* subspecies) presence in one area in the SE tip of the country ([33]; Figure 1), the Sidi Toui National Park, a 63 km$^2$ protected area effectively excluding livestock grazing by a full perimeter double fence that was not reached by our survey, since it is located out of the reported range in Tunisia [18].

Finally, data obtained from the E-Bird platform are summarized in Table 2. In the period after our survey (2007), Dupont´s larks have been reported in Tunisia for just two sites—one coastal site even more northern than Kasserine (Mahdia) and the other in the southern part of the country (Ksar Ghilane, not reached by our survey; Figure 1). Sightings of Dupont´s lark were widely reported in the E-Bird platform for Spain and Morocco, even with a clear increasing trend in the frequency of sightings when comparing the periods before and after 2007 (Table 2). In contrast, reports of the species for the rest of Maghreb were extremely scarce and deeply declining after 2007 (Table 2). Sightings of Dupont´s lark within the supposed range of the margaritae subspecies were even more scarce (18 birds in 9 sites), particularly after 2007 (just two birds in two sites—Ksar Ghilane in southern Tunisia and one in Egypt; Figure 1 and Table 2).

**Table 2.** Summary of Dupont´s lark observations uploaded to the E-Bird platform (revised on 10 January 2023). Data split by date as 2007 and years before and by years after 2007.

|  | Tunisia | Morocco | Algeria | Libya | Egypt | Spain |
|---|---|---|---|---|---|---|
| IN OR BEFORE 2007 |  |  |  |  |  |  |
| No. of points with observations | 3 | 29 | 7 | 1 | 2 | 165 |
| Total number of birds | 6 | 48 | 8 | 6 | 2 | 490 |
| Average number of birds per observation | 2 | 1.6 | 1.1 | 6 | 1 | 3 |
| AFTER 2007 |  |  |  |  |  |  |
| No. of points with observations | 2 | 45 | 0 | 0 | 1 | 599 |
| Total number of birds | 2 | 94 | 0 | 0 | 1 | 3658 |
| Average number of birds per observation | 1 | 2.1 | 0 | 0 | 1 | 6.1 |
| Most recent estimated extent of ocurrence (km$^2$) | <100 [1] | 1645 [2] | 1200 [3] | 1000 [3] | 200 [3] | 1010 [4] |
| Estimated population (singing males) | <600 [1] | 15,400 [2] | – | – | – | 3800 [4] |

[1] This work, Birdlife International 2023; [2] García et al. 2008a, Birdlife International 2023; [3] Birdlife International 2023; [4] Traba et al. 2021.

## 4. Discussion

Our search effort in Tunisia was relatively low as compared with Morocco (125 playback trials vs. 530 for study areas of similar extent, see [7]). However, using the same methodology, we found frequencies of detection of Dupont´s lark and population densities in alfa stands only slightly higher in Morocco (0.58 birds/trial) than in Tunisia (0.45), while detection frequency was clearly higher in other habitats of Morocco (0.36 birds/trial vs. no birds detected in 45 trials in Tunisia), which suggests that the absence of records of the species in other habitats of Tunisia is a reliable result. Overall, three main conclusions may be highlighted, coincident with research performed in Morocco [7,8,22], and can thus also likely be extended to northern Africa as a whole: (1) the species range has greatly reduced during the last decades with respect to potential distribution estimated from older data, particularly in the southern parts of the original distribution; (2) the species population size is very small, particularly for a passerine, and that may have also been reduced in recent times; and (3) the species is intimately linked to alfa steppes, even more than what was reported from Morocco (Table S1, [7]), and a deep regression and degradation of this habitat is most likely the main factor behind the decline of the bird.

Current distribution of Dupont's lark in Tunisia seems to be different from and much more limited than the most recent estimate [18]. In fact, a good deal of the core area we have detected is partly out of the northern boundaries reported by those authors. This distribution has adjusted to the current extent of well-preserved alfa grasslands, as observed in Morocco [7]. In the Gafsa region, at the southern edge of the core distribution area that we have detected, as well as in the rest of the potential range that we have sampled (Tataouine, Ben Guerdane, Douz, Kebili), alfa grasslands have already disappeared or are markedly degraded, almost always with a vegetation cover smaller than 5%, clearly below the optimal range for Dupont's lark [6,7]. To our knowledge, there are no recent records of Dupont's lark in the Gafsa region, where we were unable to find it as well. In Tataouine and Douz-Kebili, there are some recent records during the breeding and post-breeding season (references in [18]). However, in these regions, the observations involving several individuals during the breeding season are old, from the 1930s, while almost all observations reported later than the 1980s were only a few individuals (1 in Tataouine-Ben Guerdane, 3 in Kebili), suggesting that this was an already rare species by that recent time. We have prospected intensively in those two regions (48 trials in all the habitat types and in trying to select patches with maximal quality for the species) with negative results, and this raises clear concerns about the fate of the species in this large area. However, given the huge size of these areas and our limited fieldwork, we cannot discard the notion that some relict and small populations could still survive, something typical in other areas of Spain and Morocco where the species is declining [6,7,11]. Interestingly, one of the two recent records of the species in southern Tunisia, after our survey in 2007, corresponds to a protected area with well-preserved steppe vegetation due to livestock exclusion [33]. The other recent sighting in southern Tunisia, obtained from the E-Bird platform, corresponds to the Ksar Ghilane area, an isolated oasis that also seems to have some well-preserved patches of steppe vegetation (judging from satellite images). The latter two areas were not covered by our survey (Figure 1).

Our estimate of approximately 500 singing males in Tunisia considering the mean density of the second method must be interpreted with caution due to three potential biases. Firstly, the extent of suitable habitat for the species is difficult to ascertain due to the detailed scale that is needed to appropriately reflect the relevant parameters, such as vegetation cover and composition, soil type, and slope. Our method seemed reliable enough to delimit most of the core areas for the species at large scale. However, these core areas of alfa steppes, when observed on the field at smaller scale, may have a mixture with less adequate natural vegetation, such as *Artemisia campestris* and *A. herba-alba* or other shrubs (e.g., *Haloxylon articulata*, *Astragalus armatus*, etc.), or even with recent plantations of *Eucaliptus* spp. or *Opuntia* spp.—all of them cases that would have remained undetected

in the satellite images that we used. Dupont's lark densities in optimal alfa steppes of Tunisia were similar to those found in Morocco, but densities in mixed habitats between alfa and other shrubs should be smaller [7]. We have applied the Dupont's lark density of optimal alfa habitat to a total area where the real extent of this habitat would be lower, and consequently, overall population would be overestimated. In fact, the population estimated by the other method, intensively sampling bird presence in two Tunisian areas that included optimal alfa patches, but also some patches of suboptimal habitat with Opuntia and other shrubs, would be smaller than 400 birds.

Secondly, as detailed previously, we cannot discard the existence of some small remnant population to the south of the core area that we detected (as supported by the two recent sightings), and this would increase our population estimate. In any case, populations should be very small in these relict sites since no birds were detected at all with a sampling effort only slightly lower than the one spent in the core northern area, and the habitat prospected in the southern areas during our field trip seemed poorly suited to the species, even though we selected the best habitat patches found for our trials. Thus, the possible increase in our population estimates that could be caused by these undetected hypothetical southern populations would not in any way change the worrying status of the species we are presenting. Finally, our population estimates depend on the quality of our methodology, which, due to the sampling dates, was not based on the most widely accepted method for this species (early morning census of singing males). However, both methods that we used to estimate population density provided results of similar magnitude, within a reasonable error range. Both estimates are remarkably similar if we take into account that the density obtained from intensive searching in parcels must be considered an underestimate and that the minimum value of the 95% confidence interval of the density calculated from the comparison with Morocco is practically the same number. In summary, even with a certain degree of uncertainty, the population in the core area that we have detected should be lower than 600 territorial males, most likely between 350 and 500, and the most optimistic estimate for the whole country would not in any case exceed 800–1000 males. These figures are much lower than estimates from Spain (3700–4000 singing males [14]) and Morocco (15,400 singing males [7]); therefore, the Tunisian population must be considered the most endangered of those for which population estimates have been provided—something particularly worrying given the genetic and phenotypic singularity of this population [8]. Thus, the situation of the species in Tunisia should be considered critical since the overall estimated population is likely much lower than 1000 birds, its range seems to be largely reduced, and its overall current extent of occurrence, including the main core area in Feriana-Kasserine with a high density of the species, is no more than 70 km$^2$. Finally, it is remarkable that all the birds that we were able to detect were found on alfa grasslands or mixed habitats with alfa grasses and xerophytic shrubs dominated by the former, a pattern of habitat selection similar to that found in Morocco [7] (Table S1). If something, Dupont's lark in Tunisia seems to be even more restricted to alfa grasslands, and this raises interesting questions about the evolutionary origin and biogeographical distribution of this variation in habitat selection patterns.

The concern about the status of the species must be raised even higher when considering the particular taxonomic status of the species in Tunisia, where the two described subspecies, *C. duponti duponti* and *C. duponti margaritae*, should exist. According to [18] and [24], the former would occupy the northern part of the range in Tunisia, likely connected with populations in Algeria and Morocco, while the latter would extend over the more southern and arid part of the potential range, also related with the scarce Libyan, Egyptian, and southern Algerian populations [10]. The latter is precisely the area of Tunisia where the species was not detected, where habitat was more degraded, and where the only two recent sightings of the species support that *margaritae* in this area may be surviving only in the few and small patches with well-preserved steppe vegetation. The genetic and morphological singularity of the Tunisian population of Dupont´s lark has been recently confirmed [8]. Both individuals, genetically related with the population in Morocco and

birds with phenotypic traits corresponding to the subspecies *margaritae* and haplotypes undetected in Spain or Morocco, have been found in the same population in the north of Tunisia [8], where hybridization between both subspecies has also been confirmed recently [23]. Thus, the subspecies *margaritae* could have disappeared from most of its original range in Tunisia due to habitat degradation. Furthermore, at least some of the populations originally occupying southern Tunisia would have moved to the north where they are now genetically mixing with the nominal subspecies, which would increase the concern about the conservation of this taxa. The results obtained from the E-Bird platform also support that the status of *margaritae* may be critical over the whole original range of the subspecies. Sightings of Dupont´s lark in Spain and Morocco were relatively abundant and increasing after 2007, likely due to the increasing popularity and use of the E-Bird platform by birders around the world, given that the actual abundance of the species has been declining over last half century in both countries [7,14]. In contrast, observations within the range of *margaritae* were extremely scarce and declining, with only three sites with sightings after 2007 (Figure 1, Table 2). The absence of recent records from Algeria and Libya may be related to few birders, and tourists in general, visiting those two countries with conflictive social, economic, and political conditions making them unattractive for tourism. However, this would not be the case for Tunisia and Egypt, which are major tourism and birding sites in North Africa, including Morocco. In summary, all available information supports the notion that the status of *margaritae* subspecies is most likely critical over the whole range (see additional information for the case of Algeria in [23]). Additional surveys, including genetic work, using the most accurate and efficient methodology to detect this elusive species [34] are urgently required in the Maghreb beyond Morocco in order to set the basic scientific framework necessary to secure conservation of this taxa, which is likely critically endangered at this current time.

*Remarks for Conservation*

As in the case of Morocco [7,22], the main conservation problem that we have detected for Dupont´s lark in Tunisia is the disappearance or degradation of its main habitat—alfa steppes. In this sense, the recent regression of Dupont´s lark to northern Morocco and Tunisia, could be, at least partially, a consequence of global warming affecting the habitat of a species with a highly selective pattern of habitat selection—one of the main expected effects of global warming on animals [1,2,35]. Increased overgrazing is also of concern, since it is considered the main threat for natural vegetation in northern Africa [5,36] and to which alfa steppes are particularly sensitive [27,37]. However, in the case of Tunisia, agricultural development affecting natural vegetation areas seems to be particularly important ([18]; authors pers. obs.). In fact, we could observe recent new plantations of olive groves or cereal on alfa steppe clearings in the areas with the highest densities of Dupont´s lark, likely in patches with good soil and hydric conditions, and thus, those more favorable for future maintenance of alfa patches under a scenario of global warming. Even more paradoxical, the fight against desertification is also a current enemy of alfa steppes in Tunisia since there is an ongoing program of *Eucaliptus* spp. and cacti *Opuntia* spp. plantations all over the country, affecting the alfa steppes with Dupont's lark. Tunisia is considered to be the North African country most intensely involved in the recent fight against Saharan advance [5]—an effort that must be welcomed and encouraged. However, we strongly suggest that this necessary and advisable program carefully choose the areas where it is applied, preserving the quality of a vegetated habitat as important for species conservation as the alfa steppe is, not only for Dupont´s lark but also for other species of delicate conservation status in northern Africa, or even at global level, such as the Pin-tailed Sandgrouse (*Pterocles alchata*), or the Scrub Warbler (*Scotocerca inquieta*). Interestingly, one of the few recent proofs of presence of Dupont´s lark in southern Tunisia, at Sidi Toui National Park, corresponds to a protected area that has as the major aim of protecting fragile sub-desert vegetation by excluding livestock grazing; it is a sort of island with well-preserved low-shrub vegetation adequate for Dupont´s lark surrounded by desertified

steppes inadequate for the species ([33]; pers. obs, during field work in nearby areas) highlighting the importance of well-managed protected areas preserving steppe vegetation for the conservation of last populations of this species at the desert edge.

Conservation of steppe habitats for Dupont's lark has been recently considered a priority in Spain [14,31], even though in the mid to long-term, natural vegetation areas are expected to recover in that area [27]. In this respect, the situation in North Africa must be considered more worrying since Dupont´s lark habitat selection there is stricter, its main habitat is quickly disappearing, and the process is still running at a good pace [27]. Degradation of alfa grasslands may be particularly difficult to solve since the degradation to shrub steppe of other species is considered an irreversible process. Moreover, even if stressors causing degradation are relieved [27], direct management to recover alfa grasslands would still be necessary. Given the worrying situation of the species we are reporting, action is urgently required to stop habitat loss.

Finally, we would also like to report which strategy could be the main one to preserve the remaining Tunisian alfa steppes, based on a factor likely explaining the current presence of good patches of habitat in the Feriana-Kasserine region: the involvement of local rural people to preserve it and interested to the point of maintaining alfa cultivation. Reports by local people allow us to affirm that: (1) the best alfa grasslands we have found in Tunisia are the ones exploited as a source of cellulose for the big "alfa-paper" factory of Kasserine (said to be a high quality paper); (2) alfa harvesting is an important source of income for the rural population in a huge area from August up to January; and (3) alfa harvesting could be nicely fitted to Dupont's lark maintenance since it is developed outside of the breeding season. We suspect that the future of this relict population may be strongly linked to the fate of that factory, although the overharvesting of alfa grass would have to be controlled as well [27].

**Supplementary Materials:** The following supporting information can be downloaded at: https://www.mdpi.com/article/10.3390/d15040549/s1, Figure S1: Current estimated distribution of Dupont's lark in the Maghreb (shaded areas) derived from Birdlife International data zone (http://datazone.birdlife.org/species/requestdis). Distribution within Tunisia was estimated taking into account data presented in this paper (mentioned as Suarez et al. *in litt*); Table S1: Number of broadcasting trials in different habitat types of Tunisia (most common species in the patch). Dupont´s larks were detected exclusively in alfa grass. In Morocco, 85,2% of detections occurred in pure or mixed patches of alfa grass, 9.5% in *Noaea* spp. and 5.3% in *Artemisia* spp. [7].

**Author Contributions:** Conceptualization, F.S.; methodology, J.T.G. and F.S.; software, F.S.; validation, all authors; formal analysis, F.S.; investigation, all authors; resources, all authors; data curation, F.S.; writing—original draft preparation, F.S.; final writing—review and editing, J.V. and J.T.G. All authors have read and agreed to the published version of the manuscript (except F.S., who unfortunately passed away before reading this final version).

**Funding:** This research received no external funding.

**Data Availability Statement:** The data presented in this study are available upon request from the corresponding author. The data are not publicly available due to privacy restrictions as it is being used in further studies for upcoming publications.

**Acknowledgments:** To A.P. Ozenda, for his excellent book about Maghreb vegetation, that relieved our souls for many days of unfruitful searches of larks. We would also like to thank all the people who participated in the Moroccan expeditions: V. Garza, J. Hernández, A. Ramírez, J. Oñate, I. Hervás, R. del Pozo, E. Ramírez. M. Calero-Riestra, and E.L. García. This study is dedicated to the memory of our beloved colleague Quico Suárez, hoping that this version could have been satisfactory for him.

**Conflicts of Interest:** The authors declare no conflict of interest.

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
