# Peer review of "Marked Range Regression and Possible Alteration of Distribution of the Dupont’s Lark Chersophilus duponti in Tunisia: Conservation Consequences of Vanishing Alfa Grass Stipa tenacissima Steppes in North Africa"

_diversity, doi:10.3390/d15040549_

Round 1
Reviewer 1 Report
Diversity-2243598
This paper reports on a survey of Dupont’s lark in central Tunisia and compares the results to other regions from the birds range, comments on it conservation status and discusses future action for its conservation. It is a well considered paper reporting on the status of species in need of conservation efforts. I recommend the paper be published after addressing the following comments.
General Comments
Goals – The second goal of the paper is to ‘characterize its preferred habitat and estimate its population size’. However, throughout the paper, the authors refer to definitions of preferred habitat for sampling method description and area of occurrence definitions, and do not actually outline any method for estimating preferred habitat. I suggest rewording this goal such that habitat is stated to be known and used for the current work, rather than unknown and in need of characterization.
The methods for estimating occupancy require additional description. There is a description of patch definition and site selection where call-back detections were used. However, there is no reporting of overall effort, how many sites? How many sites per patch? How much coverage per area in each patch? Were the sites spatially independent? Without knowing these methods, it is difficult to understand what the patch occupancy means. For example, a single detection in a very large patch (all patches were > 1 km2) may correspond to an ephemeral occurrence that is not representative of the actual breeding population. A map with the sites is needed.
It can be assumed from the description that there was no repeat sampling at each sites, and as such there is no estimate of detectability, which likely biased the occupancy estimate low.
Inline comments
Abstract – The methods description here is confusing. Was the satellite imagery used to fit a model based on ground observations of occurrence? How was the suitable habitat defined and estimated for the km2 cover metric (line 19)?
35 – gr. change ‘…to the north..’ to ‘northward’
61 – gr. ‘thought to’
63 – sp. ‘nominate’
68 – gr. ‘would be’
61 – the description of the sub-species and possible uniqueness of the Dupont’s lark in Tunisia in this paragraph is compelling. The paragraph could be improved by emphasizing near the end that this uniqueness is additional motivation for estimating the habitat quality, distribution and population of the lark in Tunisia, even though it may not be clear which subspecies is necessarily being sampled.
Figure 1 - An additional inset showing a broader scale context of Tunisia would be helpful for placing the study area more thoroughly. In the figure legend, how are occurrence and occupancy distinguished? This map also needs a legend. It is not clear what the symbology means.
134 – ‘min’ not ‘m’
138 – it is not clear how patch occupancy was defined. Was any single detection in any patch size sufficient to define a patch as occupied?
150 – because the removed patches in the definition of area of occurrence were simply those patches with vegetation types with no detections, a far more thorough description of patch types is needed, preferably in a table. Also, patches with ‘no detections’ may be occupied, but with no estimate of detectability, this is an undefined metric.
168 – Relying on Moroccan density estimates seems a bit shaky, given that the authors outline the distinctiveness, morphological, genetic, of the Tunisia population. A thorough description (table) of habitats, and how they compare, in character and use by the lark, between the two countries would provide additional confidence here.
209 – what is meant by ‘correcting for the small difference in response frequencies’?
284 – gr. ‘all of these cases…’
387 – Are there any data from Sidi Toui National Park, from this study?
396 – gr. …’ it’s main habitat is disappearing…’
399 – sp. ‘shrub’
Author Response
We would like to thank reviewer1 for his work and good suggestions to improve the paper. We provide a response to his criticisms below (in italics)
REVIEWER 1
This paper reports on a survey of Dupont’s lark in central Tunisia and compares the results to other regions from the birds range, comments on it conservation status and discusses future action for its conservation. It is a well considered paper reporting on the status of species in need of conservation efforts. I recommend the paper be published after addressing the following comments.
Thank you for your positive consideration of our work. We have followed your kind advise over our draft below.
General Comments
Goals – The second goal of the paper is to ‘characterize its preferred habitat and estimate its population size’. However, throughout the paper, the authors refer to definitions of preferred habitat for sampling method description and area of occurrence definitions, and do not actually outline any method for estimating preferred habitat. I suggest rewording this goal such that habitat is stated to be known and used for the current work, rather than unknown and in need of characterization.
The true situation when we were planning field work in Tunisia is that in that country the species seemed to occupy “high Artemisia steppe and alfa (Stipa) fields on hard or pebbly soils”, following the main available reference in that country, the book by Isenmann et al. 2005. However, by that time it was already clear that in Spain the species occupied a much more varied range of habitats with respect to dominant vegetation, and that structural factors (flat or slightly undulated terrain, hard soil, high vegetation cover of relatively low height) were more important than the shrub species dominating the occupied vegetation community (Garza & Suárez 1990; Traba et al. 2021). Thus, we decided to sample Artemisia and Stipa steppes, but also other vegetation types with the adequate structure (as reported in the first paragraph of Methods) to avoid any possible overlooking of the species due to biased habitat sampling. In this way, we could be sure that our survey was developed on every habitat potentially adequate for the species (we provide now a full report of habitats sampled, as compared to Morocco in Table S1). Then, the reality we found over the terrain is that there were really few vegetation patches with adequate structure for the species, so we sampled all the few patches we detected with well preserved alpha grass, but also of other vegetation types, finding the species just in alpha grass (Table 1). However, the referee is right when pointing out that we really did not make specific work about habitat selection, so we have reworded that goal in the introduction as “sampling all potential current habitats for the species to provide an unbiased estimate of its distribution and population size”, that is a better definition or our aim.
The methods for estimating occupancy require additional description. There is a description of patch definition and site selection where call-back detections were used. However, there is no reporting of overall effort, how many sites? How many sites per patch? How much coverage per area in each patch? Were the sites spatially independent? Without knowing these methods, it is difficult to understand what the patch occupancy means. For example, a single detection in a very large patch (all patches were > 1 km2) may correspond to an ephemeral occurrence that is not representative of the actual breeding population. A map with the sites is needed.
We have included additional information about our survey to satisfy this request, including more details about the survey in Fig. 1, some improvements in the methods section in the line of reviewer’s requests and Table S1 showing with more detail of all the habitats sampled. We hope all this can satisfy this reviewer’s caveat, allowing any possible future replication of our survey, but we are ready to make additional changes if this would not be enough for his judgement. We would like to clarify that the overall aim of our expedition to Tunisia was not only to update knowledge about distribution and abundance of the species, but also to capture birds to obtain information useful to distinguish between subspecies (pictures, measurements and blood samples for genetic analyses, information that will be presented in other related paper mentioned in the text as submitted, reference 23). Thus, in the only area where the species was detected we made an intensive searching and captured several birds (now detailed in Fig. 1). In this way, there was no any occupied patch where we detected just one bird and minimal number of broadcasting trials was 2 (now detailed in methods). In fact, this is a typical situation for the species, both in Spain and Morocco: whenever you detect an individual in a given patch of good habitat, surely there will be more in the same patch, because this is a highly resident species with strict habitat requirements, so it is common finding them concentrated in good habitat, and this should be particularly expected when habitat degradation is rampaging, as in the case of Tunisia. Furthermore, this is a species with strong territorial behaviour, so it would be very strange finding an “ephemeral occurrence” of such an species with the method used, based on territorial birds responding to a broadcasted song (it is not expected that individuals not holding a territory, e.g. dispersing birds, would approach such a call, potentially dangerous for them). Finally, it is important to highlight that our survey was developed near or already in breeding season in North Africa, that may be start as soon as February, with first clutches in March, so we also expected high response to the broadcasting trials (see references 7, 8 and 18).
It can be assumed from the description that there was no repeat sampling at each sites, and as such there is no estimate of detectability, which likely biased the occupancy estimate low.
As it was reported in the methods, we sampled the few good habitat patches we found and we used to make more than one broadcast spaced enough to be a different territory whenever the patch was large enough (we have included new details about this in Fig. 1l), but we did not repeat sampling at any site due to logistic and time constraints. However, and as it was reported in the discussion, this method had been previously tested in Morocco (where ecological conditions for the species are similar) providing reliable estimates. We agree that with our limited survey we cannot provide a full vision of current distribution and numbers of the species, our results surely provide minimal estimates. In fact, we recognised in the discussion that our limited survey only could provide a minimal estimate both of distribution and numbers, and we provided some rough maximal estimates taking into account this limitation. Finally, we recognised that more work is necessary now to identify the full distribution, particularly given the recent appearance of records out of the area we surveyed (and also out of the distribution area estimated by Isenmann et al. 2005).
Inline comments
Abstract – The methods description here is confusing. Was the satellite imagery used to fit a model based on ground observations of occurrence? How was the suitable habitat defined and estimated for the km2 cover metric (line 19)?
We did not fit any model, but we used instead the simple methodology and calculations recommended by IUCN to estimate extent of occurrence and area of occupancy, exactly the same that was done with data from Morocco (see reference 7). We have rewritten this part of the abstract trying to avoid confusions and reflect better what we have done and our results. We think that adding more details about the methods would not be appropriate for an abstract and would extend the text too much for this (by definition) short section, but we are prepared to make more changes in the reviewer or the editor think this is not enough.
35 – gr. change ‘…to the north..’ to ‘northward’
Done
61 – gr. ‘thought to’
Done
63 – sp. ‘nominate’
Done
68 – gr. ‘would be’
Done
61 – the description of the sub-species and possible uniqueness of the Dupont’s lark in Tunisia in this paragraph is compelling. The paragraph could be improved by emphasizing near the end that this uniqueness is additional motivation for estimating the habitat quality, distribution and population of the lark in Tunisia, even though it may not be clear which subspecies is necessarily being sampled.
We have added a sentence in this paragraph, as suggested.
Figure 1 - An additional inset showing a broader scale context of Tunisia would be helpful for placing the study area more thoroughly. In the figure legend, how are occurrence and occupancy distinguished? This map also needs a legend. It is not clear what the symbology means.
We have added a map in Supplementary Material showing current known distribution of the species as reported by IUCN (Fig. S1). Furthermore, we have made many changes to Fig. 1 and in the main text with the aim of providing more details about our survey, with the corresponding changes to the Figure caption, taking into account these comments.
134 – ‘min’ not ‘m’
Done
138 – it is not clear how patch occupancy was defined. Was any single detection in any patch size sufficient to define a patch as occupied?
As reported above (and now also in Fig 1C), there was no any patch with a single detection.
150 – because the removed patches in the definition of area of occurrence were simply those patches with vegetation types with no detections, a far more thorough description of patch types is needed, preferably in a table. Also, patches with ‘no detections’ may be occupied, but with no estimate of detectability, this is an undefined metric.
We have added an additional Table with details of all habitats sampled as Table S1. We have specified in this paragraph (and in new Fig. 1) that we calculated area of occupancy and extent of occurrence just for the only area where we detected the species (Feriana-Kasserine region). I guess this could be the source of confusion, because we did not remove any patch in the definition of area of occurrence, but we assumed that patches of the same habitat within a radius of 30 Km to the patches where we found the species could also be occupied by the species (extent of occurrence). We think that new Fig. 1 will also help to avoid confusions.
168 – Relying on Moroccan density estimates seems a bit shaky, given that the authors outline the distinctiveness, morphological, genetic, of the Tunisia population. A thorough description (table) of habitats, and how they compare, in character and use by the lark, between the two countries would provide additional confidence here.
A short text has been included in the caption of Table S1, reporting the habitats sampled in Morocco for comparative purposes. We do not include more details because this information has been published elsewhere (reference 7). We would like to clarify that the uniqueness of Dupont´s larks in Tunisia, refers exclusively to the subspecies margaritae. But as reported in the discussion, the birds found in Feriana-Kasserine were apparently of both subspecies occupying the same habitat and area (and now we know that with ongoing hybridization, reference 23). That is, the area for which we estimate density with a method based on data from Morocco also holds the same subspecies than in Morocco, under pretty similar environmental and ecological conditions, what justifies using those estimates from Morocco. An additional result supporting the use of data from Morocco is that the rate of positive trials in alfa grass stands was very similar in both countries (Tunisia, 41.3 %, n = 63; Morocco, 42.1 %; as reported in the text, see next comment). This strongly support that bird density in alpha patches of both countries was quite similar. In any case, this is just one of two methods considered, the difference between estimates provided by both methods were not too high, and minimum estimate provided by estimates obtained in Morocco were very similar to the estimate obtained by the other method.
209 – what is meant by ‘correcting for the small difference in response frequencies’?
We refer to the small difference in percentage of positive trials reported in the previous sentence. We have reworded this sentence to clarify this.
284 – gr. ‘all of these cases…’
Done
387 – Are there any data from Sidi Toui National Park, from this study?
Not really, our survey did not reach that southern area that was out of the estimated distribution reported by Isenmann et al. 2005, because we attached our survey to that known distribution area, as reported in the methods section and Fig 1. In any case, we have clarified now this in the text with a short sentence.
396 – gr. …’ it’s main habitat is disappearing…’
Done
399 – sp. ‘shrub’
Done
Reviewer 2 Report
The comparison between the present (2007) suitable habitat surface and effective area of occupancy and the former distribution in Tunisia is not acceptable. It must be removed from the abstract. These two studies have been completed using different methodologies. The authors of the monography from 2005 had not used satellite images and GIS programs for the period of their studies, which included all avifauna of Tunisia.
In my opinion, using the ebird data should be done with caution. Many of the observations there are uploaded by unexperienced birders. Furthermore, the general picture in ebird maps rarely is consistent with the reality. Thus conclusions made on the basis of ebird alone must be avoided. Thus I cannot understand how the inclusion of these data as a table in the manuscript contributes to the aims of the study especially taking into account the scarce data from Tunisia.
Author Response
We would like to thank reviewer 2 for his work and good suggestions to improve the paper. We provide a response to his criticisms below (in italics).
The comparison between the present (2007) suitable habitat surface and effective area of occupancy and the former distribution in Tunisia is not acceptable. It must be removed from the abstract. These two studies have been completed using different methodologies. The authors of the monography from 2005 had not used satellite images and GIS programs for the period of their studies, which included all avifauna of Tunisia.
We do not understand why these contrasting distributions should be removed from the abstract, given that this is a major conclusion of the paper. The methodology used by Isenmann et al. was the classical work of revision of previous known ornithological information, plus compiling any known recent records of the species, as commonly used in this kind of bird guides for a given country. We have just made a more specific sampling for one of the species included in that guide and compare the results, we can not see the problem here. Furthermore, as we reported in the text, the information considered by Isenmann et al. 2005 for this species was quite old, with very few recent records reported (particularly when compared with other more common or easy to detect species in Tunisia). Probably this information was already supporting that this species was becoming more rare in the country than previously known by the time these authors wrote the book (not much earlier than our survey), but they did not have more precise results to change the known distribution of the species, and made what was correct (do not change a distribution map if you do not have enough information to support that change). In this sense, our work would just simply confirm with additional data what these authors were already reporting in the text for this species: “Overgrazing and agricultural development have caused a reduction in its preferred habitats and a considerable decrease in numbers”. We simply are reporting that, thanks to a deeper survey, we have detected a huge decline in numbers, probably higher than suspected by Isenmann et al. 2005, and due to the extensive degradation of habitats surviving individuals may have moved to remaining adequate habitats out of the original known distribution area in the country. We recognise at several points of the text that Isenmann et al. 2005 were already reporting that this species was suffering a big decline, acknowledging their good assessment. However, we have reworded this part of the abstract to clarify that we are comparing a “minimal extent of ocurrence and area of occupancy” derived from our survey with “presence of the species in adequate habitat over ca. 89 000 km2”, as reported by Isenmann et al. 2005. We think these are more fair and precise definitions of what we are comparing here.
In my opinion, using the ebird data should be done with caution. Many of the observations there are uploaded by unexperienced birders. Furthermore, the general picture in ebird maps rarely is consistent with the reality. Thus conclusions made on the basis of ebird alone must be avoided. Thus I cannot understand how the inclusion of these data as a table in the manuscript contributes to the aims of the study especially taking into account the scarce data from Tunisia.
We completely agree that e-bird data must be used with caution, but we think we have applied that caution to our analysis and discussion of those results. First of all, judging from observations in Spain and Morocco, a high proportion of observations of this species were reported by professional ornithologists and experienced birders (the name of observers is provided for each observation and we know most people in the little world of professional ornithologist and experienced birders in both countries). This is not strange for this elusive species that rarely would be detected or identified by inexperienced birders. Of course e-bird cannot be considered the reality and we have not extracted any conclusion made on the basis of e-bird alone. What we have made is what we think is a pretty interesting comparison between the evolution in the frequency of observation of this species in different countries, fully discuss this result, considering potential bias or mistakes, and use it as complementary information to our survey. The scarce records for Dupont´s lark in Tunisia, Algeria and Egypt as compared to Morocco and Spain are exactly the interesting point we are rising: while in Morocco or Spain the number or records of Dupont´s lark is raising last years, despite known population declines of the species (probably due to the increase in popularity and use of this new app), in the rest of north African countries the frequency of observation of this species is deeply declining, what could be explained by the same process that our survey in Tunisia is reporting: extensive habitat degradation causing a huge decline in numbers and marked reduction in distribution area. This complementary result is thus in line with main results of the paper coming from our survey, and extends our concern for Tunisian populations of the species to all eastern and southern part of North Africa where rare margaritae subspecies is supposed to live.
Round 2
Reviewer 1 Report
Additional proof-reading for flow and readability, particularly regarding long paragraphs and run-on sentences, is needed.
Fig 1. - The map looks great. However, I recommend adding a legend and reducing the caption length. The colors could be indicated in a legend so the caption is shorter.
78 - I recommend reducing the length of this sentence or breaking it into two. Currently it is a bit difficult to read due to length.
129 - This paragraph should be broken up for readability.
186 - I believe this should read ‘The ‘area of occupancy’ was considered to be the sum of the area of all those patches in which the species was recorded…’
Author Response
We would like to thank the last comments by this referee and his overall detailed revision of our manuscript. We response below in italics to those comments
Additional proof-reading for flow and readability, particularly regarding long paragraphs and run-on sentences, is needed.
We have revised again the manuscript trying to improve readability, as suggested.
Fig 1. - The map looks great. However, I recommend adding a legend and reducing the caption length. The colors could be indicated in a legend so the caption is shorter.
Done, thank you gor the good suggestion
78 - I recommend reducing the length of this sentence or breaking it into two. Currently it is a bit difficult to read due to length.
Done
129 - This paragraph should be broken up for readability.
Done
186 - I believe this should read ‘The ‘area of occupancy’ was considered to be the sum of the area of all those patches in which the species was recorded…’
Done